# Using the Traditional Ex Vivo Whole Blood Model to Discriminate Bacteria by Their Inducible Host Responses

**DOI:** 10.3390/biomedicines12040724

**Published:** 2024-03-25

**Authors:** Heather M. Chick, Megan E. Rees, Matthew L. Lewis, Lisa K. Williams, Owen Bodger, Llinos G. Harris, Steven Rushton, Thomas S. Wilkinson

**Affiliations:** 1Microbiology and Infectious Disease, Institute of Life Science, Swansea University Medical School, Swansea SA2 8PP, UK; h.m.chick@swansea.ac.uk (H.M.C.); 920430@swansea.ac.uk (M.E.R.); 794305@swansea.ac.uk (M.L.L.); lisa.williams2@hartpury.ac.uk (L.K.W.); l.g.harris@swansea.ac.uk (L.G.H.); 2Department of Animal and Agriculture, Hartpury University, Hartpury, Gloucestershire GL19 3BE, UK; 3Patient and Population Health an Informatics Research, Swansea University Medical School, Swansea SA2 8PP, UK; o.bodger@swansea.ac.uk; 4School of Natural and Environmental Sciences, Newcastle University, Newcastle upon Tyne NE1 7RU, UK; steven.rushton@newcastle.ac.uk

**Keywords:** ex vivo whole blood models, host immune responses, bacterial discrimination, pair-wise comparison, multivariate analysis, *Staphylococcus epidermidis*, *Staphylococcus aureus*, *Escherichia coli*

## Abstract

Whole blood models are rapid and versatile for determining immune responses to inflammatory and infectious stimuli, but they have not been used for bacterial discrimination. *Staphylococcus aureus*, *S. epidermidis* and *Escherichia coli* are the most common causes of invasive disease, and rapid testing strategies utilising host responses remain elusive. Currently, immune responses can only discriminate between bacterial ‘domains’ (fungi, bacteria and viruses), and very few studies can use immune responses to discriminate bacteria at the species and strain level. Here, whole blood was used to investigate the relationship between host responses and bacterial strains. Results confirmed unique temporal profiles for the 10 parameters studied: IL-6, MIP-1α, MIP-3α, IL-10, resistin, phagocytosis, S100A8, S100A8/A9, C5a and TF3. Pairwise analysis confirmed that IL-6, resistin, phagocytosis, C5a and S100A8/A9 could be used in a discrimination scheme to identify to the strain level. Linear discriminant analysis (LDA) confirmed that (i) IL-6, MIP-3α and TF3 could predict genera with 95% accuracy; (ii) IL-6, phagocytosis, resistin and TF3 could predict species at 90% accuracy and (iii) phagocytosis, S100A8 and IL-10 predicted strain at 40% accuracy. These data are important because they confirm the proof of concept that host biomarker panels could be used to identify bacterial pathogens.

## 1. Introduction

Human ex vivo whole blood models have been used for several years to investigate numerous biological, pathological and toxicological effects on immune cells in an environment that closely mimics the biological fluid present in vivo. The ability to measure soluble mediators released from cells in addition to the cell surface or functional readouts of the cells themselves confirms the utility of such models. Human blood has been used to determine inflammatory responses to pathogen-associated molecular patterns (PAMPs) [1,2,3,4] or to whole bacteria [5]. It has been used to identify viral pathogens through the IFN-γ release assay [6]. The sensitivity of human blood has found utility in assessing lipopolysaccharide (LPS) and other pyrogens, contamination of pharmaceuticals or inorganic bioaerosols [7], and is a practical surrogate for determining monocyte responses in situ [8,9,10,11,12]. We and others have used human blood to model disease processes, to compare with patient serum [13] and to uncover mechanisms of disease pathogenesis. Indeed, when combined with LPS or dexamethasone, models of immunosuppression are possible [14,15].

The three most important bacterial pathogens responsible for invasive diseases (e.g., bacteraemia and sepsis) are *Escherichia coli*, *Staphylococcus epidermidis* and *S. aureus* and are responsible for >50% of infections [16,17]. Previous studies have confirmed the pattern recognition receptors (PRRs) responsible for immune responses to these pathogens. They show that (i) *E. coli* is dependent on toll-like receptor 4 (TLR-4) [18,19]; (ii) *S. aureus* is dependent on TLR-2, NOD-like receptor (NLR) and C-type lectin receptor (CRP) signalling [20] and (iii) *S. epidermidis* induces pathways dependent [21] and independent on TLR-2 signalling [22]. In severe infections with these organisms, the diversity in inflammatory responses [23] can potentially inform on patient state and antimicrobial therapy and has the potential to offer a diagnostic tool as to the microbial source of the infection.

Current identification of these pathogens relies on traditional microbiological techniques, which take at least 24 h for identification and are vulnerable to contamination or show reduced specificity and sensitivity [24,25,26,27]. Cellular host responses, including C-reactive protein (CRP), procalcitonin (PCT), whole blood counts, erythrocyte sedimentation rate (ESR) and acute phase cytokines (TNF, IL-8, IL-6, etc.), have all been inconsistent in terms of their discriminatory power for identifying sepsis and cannot discriminate between causative organisms [28,29,30,31,32,33]. Indeed host immune signatures have only recently managed to discriminate organisms at the ‘domain’ level to separate bacterial, fungal and viral responses [34]. New model systems and proof of concept studies are needed to challenge lower order (genera, species and strain) discrimination of bacteria.

The versatility of ex vivo whole blood models suggested that they could be used to investigate the diversity of bacteria responses and used for discrimination. While it is clear that whole blood models have been applied to bacterial [35,36], fungal [37] and viral organisms [38], there are far fewer studies focusing on the potential range of whole blood responses to pathogenic stimuli in healthy donors [1]. To date, no studies have focused on whether the physiological range of ex vivo whole blood responses could be used for the discrimination of bacteria; specifically, whether whole blood responses might aid in the discrimination of bacteria at genera, species and even at strain level. This study uses an established ex vivo whole blood model [15] to generate bacterial-induced cytokine profiles of mediators associated with inflammation (IL-6, MIP-1α, MIP-3α, resistin, IL-10), coagulation (TF3), complement (C5a) and neutrophil function (S100A8, S100A8/A9 [calprotectin], phagocytosis) following stimulation with eight strains of bacteria (across three species and two genera). Then, these data were used to investigate the level of bacterial strain discrimination that could be achieved from the measured host responses.

## 2. Materials and Methods

All methods were carried out in accordance with relevant guidelines and regulations. All reagents were purchased from Sigma-Aldrich (Gillingham, Dorset, UK) unless stated otherwise.

### 2.1. Ethics

Whole blood from healthy volunteers was isolated using the vacuette blood collection system (5–9 mL) on the day of the experiment in either Lithium Heparin (Becton Dickinson, Wokingham, UK). Volunteers gave written informed consent. The project (Reference 13/WA/0190) was reviewed, and the procedures and protocols were approved by the local research ethics committee, Wales REC 6 (e-mail: wales.rec6@nhs.uk).

### 2.2. Whole Blood Infection

Three bacterial species (*E. coli*, *S. aureus* and *S. epidermidis*) were selected for profiling and were represented by 8 strains (Appendix A; [39,40,41,42,43,44,45,46,47,48,49,50,51]). One bacterial colony (of each of the 8 strains) was inoculated into a sterile broth and incubated at 37 °C overnight. The next day a standardised suspension of each bacterium was prepared in RPMI at an optical density of 0.1 OD600 nm. Then, 100 µL of standardised suspension was added to 1 mL of anticoagulated whole blood (n = 5) to a final Multiplicity of Infection of 0.2 as described previously [15,52]. The whole blood: bacteria interaction mix was incubated for 2-6 h on a Stuart SB3 rotator at 37 °C before being centrifuged at 9000× *g* at room temperature. The supernatant was removed and stored at −20 °C until ELISA analysis.

### 2.3. Membrane Cytokine Array

Cytokine screening after whole blood infection (with *E. coli* K12, *S. epidermidis* RP62A, *S. aureus* SH1000 and untreated) for 6 h was measured using the Human XL Cytokine Array kit (R&D Systems, Abingdon, UK; n = 1). The membranes were treated as per manufacturer instructions and were imaged on the BioRad ChemiDoc XRS+. Images were processed using ImageJ online software (version 1.50i; https//imagej.net/ij/index.html (accessed between January 2019 and March 2020).

### 2.4. Enzyme-Linked Immunosorbent Assay (ELISA)

The production of the proteins IL-6, C5a, TF3, IL-10, S100A8 (calgranulin A), S100A8/A9 (calprotectin), resistin, MIP-1α and MIP-3α in whole blood following infection with 8 defined strains was analysed using Duoset ELISA kits purchased from R&D Systems (Abingdon, UK; n = 5). The ELISAs were performed as per the manufacturer’s instructions, and optical densities (450 nm) were determined using a BMG Omega plate reader. Samples were blank-corrected, and a 4-parameter fit was applied to generate a standard curve from which sample concentrations were calculated (performed using BMG Mars data analysis software v3 02R2).

### 2.5. Phagocytosis

One hundred microlitres of whole blood was removed at 2, 4 and 6 h post-infection and mixed with 1.5 mL 1X FACS Lysing Solution (BD Biosciences, Wokingham, UK) before incubating at room temperature for 20 min (n = 5). Tubes were centrifuged, the supernatant discarded, and the pellet resuspended in 100 µL PBS (Thermo Fisher Scientific, Loughborough, UK). Then, 80 µL of the suspension was placed into a cytospin funnel cartridge attached to a microscope slide and centrifuged in a Cytospin 3 (ThermoShandon, Thermo Fisher Scientific, Loughborough, UK) at 300 g for 3 min. Slides were stained with Red & Purple Microscopy Hemacolor (Merck, London, UK) before being mounted with coverslips. Slides were left to dry between each described step and were viewed using a Nikon eclipse 50i microscope at x1000 magnification under oil immersion (x100 objective). Neutrophils were identified, and the number of phagocytosed and non-phagocytosed neutrophils was counted in random fields of view as previously described [52]. The percentage of phagocytosis was recorded as the degree of phagocytosis.

### 2.6. Pairwise Discrimination of Bacteria

Pairwise analysis: The process of discrimination between strains consisted of two parts, identifying the candidate discriminators and then constructing a flowchart. While most cases contained readings from 3 time points, the readings always increased with time and so only the final time point was used in each case. The most important biomarkers, and their possible use, were identified through their statistical significance and a visual inspection of the data. Once these had been chosen, a flowchart was constructed to perform a stepwise classification process using this subset of biomarkers. At each stage of the process, a simple threshold on one biomarker was used to determine which branch to follow (lower than threshold ‘turn left’ and higher than threshold ‘turn right’), until a final classification was reached.

### 2.7. Linear Discriminant Analysis of Bacteria

Individual chemokine and cytokine biomarkers were used as covariates in linear discriminant analysis (LDA) with the replicate samples of cyto-/chemokine responses to individual pathogens used as group descriptors [53]. LDA seeks to find the combination of measured biomarkers to optimise the separation of groups in the cyto-chemokine space. It is a stepwise procedure that first identifies the best discriminating variable separating groups and then progressively adds other covariates, whilst accounting for the covariation between them. Variables are only added as discriminators if they lead to a significant discriminatory effect. We used the greedywilkes function in R to identify the smallest suite of cyto-/chemokines that led to the maximum discrimination [54]. We used a three-tier hierarchy of taxonomic grouping to define our initial groups. These were strain, species and genus; the hypothesis was that we would expect the greatest discrimination using the biomarkers, firstly for genus, then species and finally strain on the assumption that the greatest differences in host responses would be greatest at the greatest taxonomical differences in pathogens. In order to assess the utility of the biomarkers for discriminating between members of each taxonomic grouping, we used the derived discriminant functions to reclassify the original data used in the LDA and then used contingency tables to illustrate the predictive accuracy of the best model for each group.

### 2.8. Data Presentation and Statistical Analysis

The data set consisted of a number of biomarker readings, with 10 different biomarkers, 8 bacterial strains, and 1 uninfected control. Preliminary analysis of the data leading to the classification flowchart was performed using ANOVA, with post hoc testing, on ln-transformed data. Logistic regression was then used to obtain the estimated thresholds controlling the branching process. Formal comparisons of the numeric differences between treatments were then performed using the (non-parametric) Kruskal–Wallis tests with Dunn’s multiple comparison tests applied. A 5% level of significance was observed. All plots display error bars denoting the standard error of the mean (SEM), and asterisks indicate the level of significance; * = *p* < 0.05, ** = *p* < 0.01, *** = *p* < 0.001, **** *p* < 0.0001. Data presentation and analysis were performed using GraphPad Prism version 7 and SPSS version 28.

## 3. Results

### 3.1. Establishment of Mediator Profiles to Use in Modelling Datasets

Building on our previous results in whole blood using *S. epidermidis* 1457 [15], eight bacterial strains (Appendix A) were used to infect whole blood prior to the determination of 10 host responses, including IL-6 (acute phase response), IL-10, MIP-1α (CCL-3), MIP-3α (CCL-20), resistin, C5a (complement activation), tissue factor-3 (TF-3; coagulation cascade), S100A8 (calgranulin A) and S100A8/A9 (calprotectin; neutrophil activation) and phagocytosis (neutrophil function). IL-6, resistin, MIP-1α and MIP-3α were identified during a cytokine microarray of whole blood factors induced by more than 3-fold following *E. coli* K12, *S. epidermidis* 1457 or *S. aureus* SH1000 infection compared to control (Appendix A). IL-8 and TNFα were also identified in this screen but were increased in response to all bacteria, suggesting poor potential for discrimination and were not studied further.

### 3.2. IL-6 and MIP Proteins Show Significant Early Induction in Escherichia Infection

IL-6 (Figure 1A–C), MIP-3α (Figure 2A–C) and MIP-1α (Figure 2D–F) levels were shown to significantly increase over the course of the infection, but with unique induction kinetics. These increases were significantly greater in the *Escherichia* genera (Figure 1A and Figure 2A,D, *p* < 0.00001) compared with both the untreated control and *Staphylococcus* genera at all time points. This trend was consistent at the species level (Figure 1B and Figure 2B,E), where *E. coli* induced significantly more IL-6 and MIP-3α than both *S. aureus* and *S. epidermidis* (*p* < 0.00001). MIP-1α was only significantly induced in *E. coli* compared to the control at the species level (Figure 2E). At the strain level (Figure 1C and Figure 2C,F), all *E. coli* strains (with the exception of strain B at 2 h) induced significant levels of IL-6 over the course of the infection compared with the control (*p* < 0.00001; Figure 1C). This trend was consistent for MIP-3α (Figure 2C) although the profile between strains differed (strains B and K12 did not induce significantly more IL-6 at 2 h and strain B did not at 4 h; Figure 1C). The profile of MIP-1α (Figure 2C–E) was considerably flatter compared to IL-6 and MIP-3. *E. coli* strains ECOR26, GMB10 and B induced significantly more than the control at 2 h only (*p* = 0.0068; Figure 2F) and strain B at 6 h only (*p* = 0.0312; Figure 2F). The anti-inflammatory cytokine IL-10 was not detected at 2 and 4 h post-infection, and no differences were found at 6 h post-infection (*p* = 0.0823; Figure 1D).

### 3.3. The Hormone Resistin Is Associated with All Infection and May Differentiate S. epidermidis and S. aureus Bacteria

ELISA analysis of resistin was performed on samples obtained at 2, 4 and 6 h post-infection (Figure 3A–C). At the genera level (Figure 3A), both *Escherichia* and *Staphylococcus* induced significantly high levels of resistin at 2 and 6 h post-infection compared with the control (*p* < 0.0001) but not significantly different from each other. At species level, it was only at 6 h that both *S. aureus* and *S. epidermidis* induced significantly higher resistin concentrations than both the control and *E. coli* (*p* = 0.0107; Figure 3B). Interactions at the strain level confirmed that only *E. coli* GMB10 induced significantly higher levels of resistin at 2 h post-infection (*p* = 0.0051; Figure 3C). At 4 and 6 h, *S. aureus* strain SH1000 and *E. coli* ECOR26 and GMB induced significantly higher levels of resistin compared with the control (*p* = 0.0012, *p* = 0.0005).

### 3.4. Neutrophils Take up Significantly More Staphylococci Than Escherichia Bacteria

Phagocytosis (%) of *S. epidermidis*, *S. aureus* and *E. coli* strains over the first 6 h of infection was used to assess the functional activity of neutrophils (Figure 3D–F). At the genera level, significantly more *Staphylococcus* was taken up by neutrophils than *Escherichia* (*p* < 0.0001; Figure 3D). No phagocytosis was detected in the uninfected controls. At the species (Figure 3E) and strain (Figure 3F) level, *S. epidermidis* 1457, RP62A and *S. aureus* SH1000 had significantly increased phagocytic responses compared to control (*p* < 0.0001; Figure 3E,F).

The neutrophil activation markers S100A8 (inactive calprotectin, Figure 4A–C) and S100A8/A9 (active calprotectin, Figure 4D–F) were found to be increased over the course of the experiment, but at no timepoint, were any significant differences between the treatment groups (*p* > 0.05). Complement (C5a) and coagulation (TF3) were also assessed over identical time courses but neither was shown to be different from the control (Appendix A). C5a was undetectable after 6 h. 

### 3.5. Pairwise Modelling Allows Bacterial Discrimination down to Strain Level

Next, pairwise relationships (Figure 5A,B) between the inflammatory parameters were investigated with the aim of developing a scheme to discriminate the bacterial strains studied (Appendix A) based on host responses generated in the ex vivo model. Reorganisation of pairwise comparisons (Figure 5A) allowed the development of a scheme with discrimination at four levels: (1) infection, (2) genera, (3) species and (4) strain (Figure 5A). When organised in this manner, increased IL-6 could discriminate between infected and uninfected samples (*p* < 0.001). Furthermore, increased IL-6 could also discriminate infection with *Escherichia* and *Staphylococcus* at the genera level (*p* < 0.001). Increased resistin showed some discrimination between *S. aureus* and *S. epidermidis* at the species level (*p* = 0.087). Similarly, increased phagocytosis could discriminate *E. coli* GMB and ECOR26 from *E. coli* strain B and K12 within a species (*p* < 0.001). In addition, increased C5a could discriminate between *E. coli* strain B and K12 at the strain level (*p* = 0.007). Finally, within the *S. aureus* species, increased S100A8/9 could discriminate SH1000 from VAP39 at the strain level (*p* < 0.001). These results confirmed the proof of concept that measurement of sufficient immune responses allows discrimination to bacterial strain level in ex vivo whole blood models.

### 3.6. Linear Discriminant Analysis Provides Host Biomarker Panels for Bacterial Discrimination

The pairwise analysis confirmed the potential of host responses for discrimination of bacteria but did not allow all inflammatory profiles to be analysed together to determine inter-relationships. Linear discriminant analysis (LDA) was used to identify a panel of inflammatory parameters that best discriminate bacteria at genera, species and strain levels (Table 1 and Table 2) and (Appendix A). In this analysis, we speculated on what level immune responses generated by a given bacterial strain could predict responses generated by strains from a given genera, species and strain. The analysis revealed that correct prediction rates of 40, 90 and 95% could be achieved for strain, species and genera, respectively, using different cyto-/chemokine biomarkers (Table 1). Furthermore, phagocytosis, S100A8 and IL-10 were the most significant discriminating immune parameters to predict strain when combined ((a) in Table 2). IL-6, phagocytosis, resistin and TF were the most significant discriminating parameters associated with species ((b) in Table 2), and IL-6, TF3 and MIP3α were the most significant discriminating parameters associated with genera ((c) in Table 2). These results further support the proof of concept that immune parameters could be used to predict the presence of individual bacteria.

## 4. Discussion

This study focused on whether ex vivo whole blood responses could be used for the discrimination of bacteria at genera, species and/or strain levels with a future goal of applying this knowledge (e.g., to bacteraemia and sepsis diagnosis). Indeed, accurate diagnosis of bacterial infections remains a significant challenge and is often hindered by variations in the immune response between different individuals. The current study used an ex vivo whole blood model to study immune responses to a core set of clinically relevant species (*S. aureus*, *S. epidermidis* and *E. coli*) that are implicated in bacteraemia and sepsis [16,34]. In addition, we sought to confirm the proof of concept of whether host responses could predict bacterial pathogens and to what discriminatory level.

Previous reports suggest that bacteria reach concentrations ranging from 1 to 10,000 cfu/mL in clinical bacteraemia with average concentrations in the 100–1000 cfu/mL range [16,55,56,57,58,59]. Considering that leukocyte concentrations are ~2 × 10^7^ cfu/mL, relevant MOIs for modelling should be in the 0.00000005–0.0005 range. However, there are a number of considerations when using MOIs in practice. Firstly, during clinical infections, leukocyte numbers and bacterial numbers change over the course of infection. Secondly, very low MOIs such as this do not stimulate early cytokine responses ex vivo because they are often rapidly cleared. Thirdly, there is now good evidence that PAMPs and DAMPs may not only be associated with live bacteria but also in soluble forms not associated with bacteria [60,61]. Fourth, viable bacteria are counted in cfu/mL, but many studies also quantify genome copies/mL using nucleic acid amplification techniques [62]. Finally, recent studies suggest that microbial composition also has a role to play in outcome measurements in experimental models of sepsis [63].

Studies to discriminate bacterial ‘Gram-status’ by their immune responses have a long history. It is clear that most bacteria induce IL-1β, TNFα, IL-2, IL-18, IL-6, IL-8, IL-12, IFN-γ and IL-10 [64,65,66,67] in blood monocytes or in whole blood models. These studies tend to show that TNFα, IL-2, IL-12 and IFN-γ responses are stronger in Gram-positive bacteria, whereas IL-6, IL-8, IL-10 and IL-18 responses tend to be stronger in Gram-negative bacteria. This is consistent with IL-6 responses observed in this study and also in Gram-negative versus Gram-positive bacteraemia [68,69,70,71]. Furthermore, others have suggested that IL-6 and IL-10 may be useful in discriminating Gram-negative from Gram-positive infections [72]. Tietze and coworkers suggest that Gram-negative species produce higher rations of IL-8/TNFα than Gram-positive species [73]. Similar experiments performed using whole animal infection in mice or fish models show strain dependence rather than species and Gram dependence and confirm the need for further strain testing [74,75]. Our results with IL-6 support clear discrimination for Gram-negative species and strains and are consistent with these studies.

The mechanism underlying these cytokine differences is still unclear; however, the different combinations of pathogen-associated molecular patterns (PAMPs) and the subsequent PRRs they activate, as well as their intracellular or extracellular localisation, could in part explain these different patterns of expression. Previous work has identified that the lipopolysaccharide (LPS) of Gram-negative *E. coli* activates TLR-4. In contrast, lipoteichoic acid [76] and lipoproteins of peptidoglycan from Gram-positive *S. aureus* activate TLR-2 and NOD-like receptors [77]. Furthermore, polysaccharide intercellular adhesin, phenol soluble modulins (PSMs) and muramylpeptides from Gram-positive *S. epidermidis* activate combinations of TLR-1, TLR-2 and TLR-6 [78,79,80]. These interactions are especially crucial when comparing the activation profile to different bacterial pathogens.

While previous studies have implicated numerous cytokines, this study identified four useful markers as bacterial discriminators: IL-6, resistin, MIP-3α and neutrophil phagocytosis. IL-6 and MIP-3α showed significant associations with Gram-negative *Escherichia coli* infections in whole blood and were consistently elevated compared with those in Gram-positive *Staphylococcus* infections. The relationship between IL-6 and MIP proteins seems intrinsically linked; in vivo, MIP-1α (and beta) has been shown to correlate with levels of IL-6 in sepsis [81]. IL-6 has already been demonstrated to be a strong predictor of severity in sepsis and with Gram-negative bacterial sepsis [82]. The close mechanistic relationship between IL-6 and MIP-3α may support the use of the latter as a diagnostic biomarker for certain types of bacterial sepsis. In support of our findings, one other study has identified increased MIP-3α in sepsis patients with diagnostic and prognostic values [83].

Resistin was identified as a potential biomarker to discriminate *S. aureus* and *S. epidermidis* at the species level. Consistent with our results, resistin has been found in elevated concentrations in the blood of patients and may be linked to an inflammatory cytokine network in the acute phase of sepsis [84]. Further studies have linked resistin to the severity of sepsis using clinical scoring outcomes [85,86]. There is less evidence for the use of resistin as a bacterial discriminatory biomarker, but our data do warrant further investigation in response to Gram-positive bacteria.

Neutrophil phagocytosis was positively associated with Gram-positive infections. This association is significant as it could have potential value in the diagnosis of staphylococcal infections. However, the disparity in the response, between Gram-positive and Gram-negative infection, could also be explained by the virulence of the strains used in the study—the *S. aureus* strains VAP39 and SH1000 and the *S. epidermidis* strains 1457 and RP62A are all known for their highly virulent phenotypes, whereas *Escherichia* strains GMB10, K12, B and ECOR26 are all regarded to be non-pathogenic strains (Appendix A). While neutrophil phagocytosis is implicated in sepsis, the exact role that bacterial ‘species’ designations play is unclear [87]. Despite this, *E. coli* strains consistently induced high levels of cytokines in whole blood, indicating a general mechanism of cytokine production, which may be responsible for decreased phagocytosis. We speculate that another reason for this disparity in phagocytosis could be associated with differences in evasion mechanisms. *E. coli* has inherent resistance for survival where iron is limited (such as in human blood and serum) by producing siderophores and phagocytosis given its colonic acid coat [88]. In contrast, *S. aureus* has less resistance to phagocytosis but tends to persist within professional phagocytes through mechanisms of resistance to reactive oxygen species and intracellular killing [89].

The complement and coagulation molecules C5a and TF3 had very specific utility in discriminating the bacteria studied. In our pairwise analysis, C5a had potential application in differentiating *E. coli* strains, B and K12. We speculate this may be related to differing types of LPS in those bacteria. In contrast, in our multivariate analysis TF3, when combined with IL-6, phagocytosis and resistin or IL-6 and MIP-3α could discriminate bacterial species and genera, respectively. This work may implicate monocyte-derived (in addition to endothelial-derived) TF3 as important in bacterial responses [90]. However, the role of C5a as a biomarker remains unclear considering the technical difficulties in their measurement due to lack of stability, its interaction with anticoagulant systems and the incomplete understanding of its receptor C5aR2 [91,92,93].

The application of pairwise and multivariate methods allowed us to confirm the proof of concept that host response signatures can discriminate between bacteria. It was clear that certain pair-wise comparisons had 100% predictive power (Figure 5B) but were limited to very well-defined comparisons, such as uninfected vs. infection or *Staphylococcus* vs. *Escherichia*. Clearly, these pairs would find utility as part of wider biomarker panels. In contrast, multivariate analysis showed a predictive power decrease in the order genera (95%) → species (90%) → strain (40%). The panels for genera (IL-6, TF3 and MIP-3α) and species (IL-6, phagocytosis, resistin and TF3) are intriguing as they overlap consistently with the pairwise analysis. It is clear that phagocytosis and resistin should be investigated further to discriminate *Escherichia*/*Staphylococcus* genera and *S. epidermidis*/*S. aureus* species, respectively.

It is vitally important that the LDA clearly identified ‘panels’ of host biomarkers as a discriminatory approach to identify bacteria at ‘genera’, ‘species’ and even ‘strain’ levels. A similar approach was used to discriminate *S. aureus*, *E. coli*, *C. albicans* and *A. fumigatus* and highlighted the importance of IL-6 and associated signalling molecules SOCS3 and IRF [37]. To date, however, studies focus on the ‘domain’ level of microorganisms, confirming host signatures for bacterial, fungal and viral responses [34]. Recently, an 8-gene panel has been used to discriminate between extracellular, intracellular and viral infections across diverse global populations [94]. Likewise, an 81-set gene signature was used to discriminate bacterial and viral infection in immunocompromised hosts [95]. However, it is clear that while domain-level discrimination using host signatures is demonstrated widely, there is a need for significant standardisation and benchmarking at the species level as cross-reactive unintended infections and aging processes can take over in the analysis [96]. It is now becoming increasingly clear that host response signatures for blood-based infection diagnostics [97,98,99] will complement similar approaches for direct detection of pathogens [100]. The application of multi-panel host-specific biomarkers to discriminate pathogens is paving the way for personalised medicine.

The use of whole blood from healthy donors allows for a comparative method of determining host response signatures for infection diagnosis. While ex vivo whole blood models have shown versatility for the study of monocyte activation [8,9,10,11,12], IFN-γ-release assays [6], cell-specific immune triggers (e.g., PAMPs) [1,2,3,4] and even the effect of storage and stability of cytokines [101,102], we believe the current application for bacterial discrimination is novel. There are significant advantages and limitations to using this model to assess healthy human responses to bacteria (Table 3). While LPS can be given ex vivo and in vivo in humans, no pre-clinical in vivo human model exists for bacterial assessment. Knowledge of the diversity of healthy human responses is relatively sparse or is only available through standardised PAMP screening [1]. Thus, expanding this model to collections of characterised bacterial pathogens is a vital next step.

We recognise that this study is not without its limitations: (i) the inclusion of eight bacterial strains across three species. While this limits the scope of the current study, it nevertheless provides sufficient data for proof of concept that bacterial genera, species and strain discrimination using the host immune response is possible; (ii) the choice of bacterial collection within a species could be better defined and include *E. coli* strains from bacteraemia patients; (iii) the relatively small number of blood donors (n = 5) is sufficient for proof of concept but does not describe the variation in host response in a healthy population [1]. Finally, it is clear that more advanced multivariate analysis and modelling approaches used to determine cytokine interdependencies and networks may also provide a rational next step for biomarker discovery [103,104,105,106].

## 5. Conclusions

In summary, this paper provides proof of concept data that host biomarker panels could be used to identify bacterial pathogens, at orders lower than ‘domain’ with increasing accuracy, strain → species → genera. Using the whole blood model as a workhorse to assay inflammatory responses but analysing them using LDA provides a simple strategy to discriminate between bacteria and the host responses they generate.

## Figures and Tables

**Figure 1 biomedicines-12-00724-f001:**
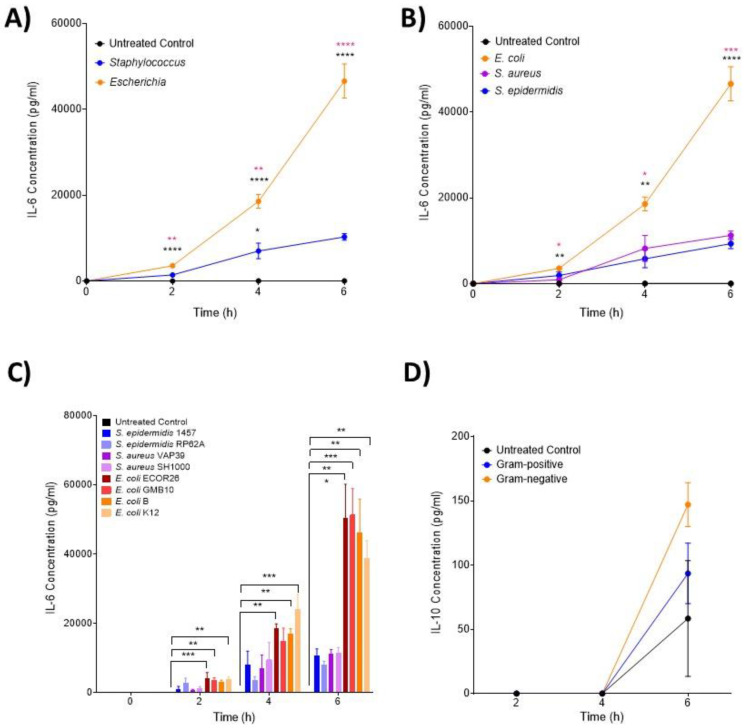
**Changes in monocyte-associated cytokines IL-6 and IL-10 following ex vivo infection**. Cytokines associated with IL-6 and IL-10 were measured by ELISA after 2, 4 and 6 h post-infection (n = 5 donors). Changes in IL-6 over time in response to (**A**) control, *Staphylococcus* and *Escherichia* genera; (**B**) control, *E. coli*, *S. aureus* and *S. epidermidis* species and (**C**) to each bacterial strain. Changes in IL-10 over time in response to (**D**) control, *Staphylococcus* and *Escherichia* genera. Error bars represent the mean ± SEM. In (**A**), black asterisk represents significant differences between the infected and control, and pink asterisk represents significant differences between *Escherichia* and *Staphylococcus*. In (**B**), black asterisk represents significant differences between *E. coli* and *S. epidermidis* and between *E. coli* and *S. aureus*. In (**B**), pink asterisk represents significant differences between *E. coli* and *S. aureus*. In (**C**), significant differences are represented between two strains and/or control by black lines and asterisks indicating the level of significance. Here, asterisks represent the following significant values: * = *p* < 0.05, ** = *p* < 0.01, *** = *p* < 0.001, **** = *p* < 0.0001.

**Figure 2 biomedicines-12-00724-f002:**
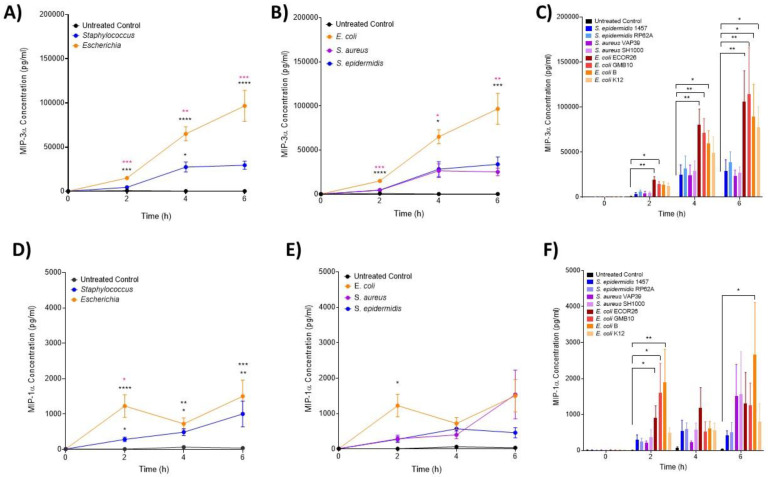
**Changes in monocyte-associated chemokines, MIP-1α and MIP-3α following ex vivo infection.** MIP-3α and MIP-1α were measured by ELISA after 2, 4 and 6 h post-infection (n = 5 donors). Changes in MIP-3α over time in response to (**A**) control, *Staphylococcus* and *Escherichia* genera; (**B**) to control, *E. coli*, *S. aureus* and *S. epidermidis* species and (**C**) to each bacterial strain. Changes in MIP-1α over time in response to (**D**) control, *Staphylococcus* and *Escherichia* genera; (**E**) to control, *E. coli*, *S. aureus* and *S. epidermidis* species and (**F**) to each bacterial strain. Error bars represent the mean ± SEM. In (**A**,**D**), black asterisk represents significant differences between the infected and control, and pink asterisk represents significant differences between *Escherichia* and *Staphylococcus*. In (**B**,**E**), * represents significant differences between *E. coli* and *S. epidermidis* and between *E. coli* and *S. aureus*. In (**C**,**F**), significant differences are represented between two strains and/or control by black lines and asterisks indicating the level of significance. Here, asterisks represent the following significant values: * = *p* < 0.05, ** = *p* < 0.01, *** = *p* < 0.001, **** = *p* < 0.0001.

**Figure 3 biomedicines-12-00724-f003:**
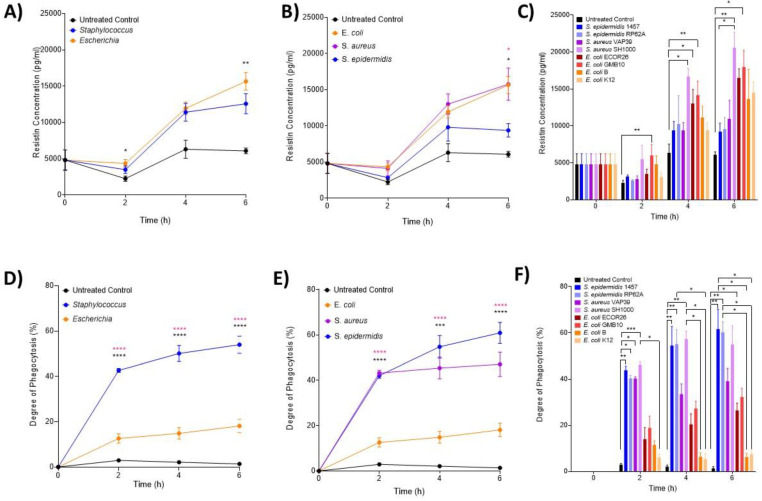
**Changes in neutrophil-associated parameter resistin and phagocytosis following ex vivo infection.** Resistin and phagocytosis were measured after 2, 4 and 6 h post-infection (n = 5 donors). Plots are presented as changes in resistin over time in response to (**A**) control, *Staphylococcus* and *Escherichia* genera; (**B**) to control, *E. coli*, *S. aureus* and *S. epidermidis* species and (**C**) to each bacterial strain. Neutrophil phagocytosis was investigated using microscopy over 6 h in response to (**D**) control, *Staphylococcus* and *Escherichia* genera; (**E**) to control, *E. coli*, *S. aureus* and *S. epidermidis* species and (**F**) to each bacterial strain. Error bars represent the mean ± SEM. In (**A**,**D**), black asterisk represents significant differences between the infected and control; and pink asterisk represents significant differences between *Escherichia* and *Staphylococcus*. In (**B**,**E**), black asterisk represents significant differences between *E. coli* and *S. aureus*, and pink asterisk represents significant differences between *E. coli* and *S. epidermis*. Finally, in (**C**,**F**), significant differences are represented between two strains or control by black lines, and asterisks indicate the level of significance. Here, asterisks represent the following significant values: * = *p* < 0.05, ** = *p* < 0.01, *** = *p* < 0.001, **** = *p* < 0.0001.

**Figure 4 biomedicines-12-00724-f004:**
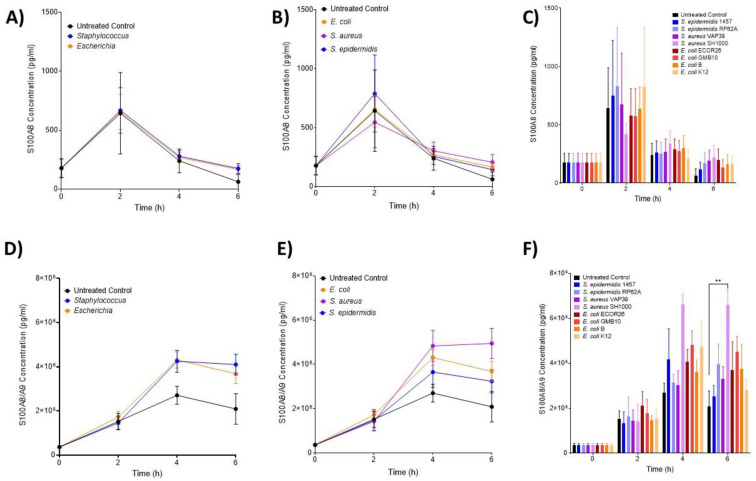
**Changes in neutrophil-associated parameters S100A8 and S100A8/A9 following ex vivo infection.** S100A8 and S100A8/A9 were measured after 2, 4 and 6 h post-infection (n = 5 donors). Plots are presented as changes in S100A8 over time in response to (**A**) control, *Staphylococcus* and *Escherichia* genera; (**B**) to control, *E. coli*, *S. aureus* and *S. epidermidis* species and (**C**) to each bacterial strain. Changes in S100A8/A9 (calprotectin) over time in response to (**D**) control, *Staphylococcus* and *Escherichia* genera; (**E**) to control, *E. coli*, *S. aureus* and *S. epidermidis* species and (**F**) to each bacterial strain. Error bars represent the mean ± SEM. Here, asterisks represent the following significant values: ** = *p* < 0.01.

**Figure 5 biomedicines-12-00724-f005:**
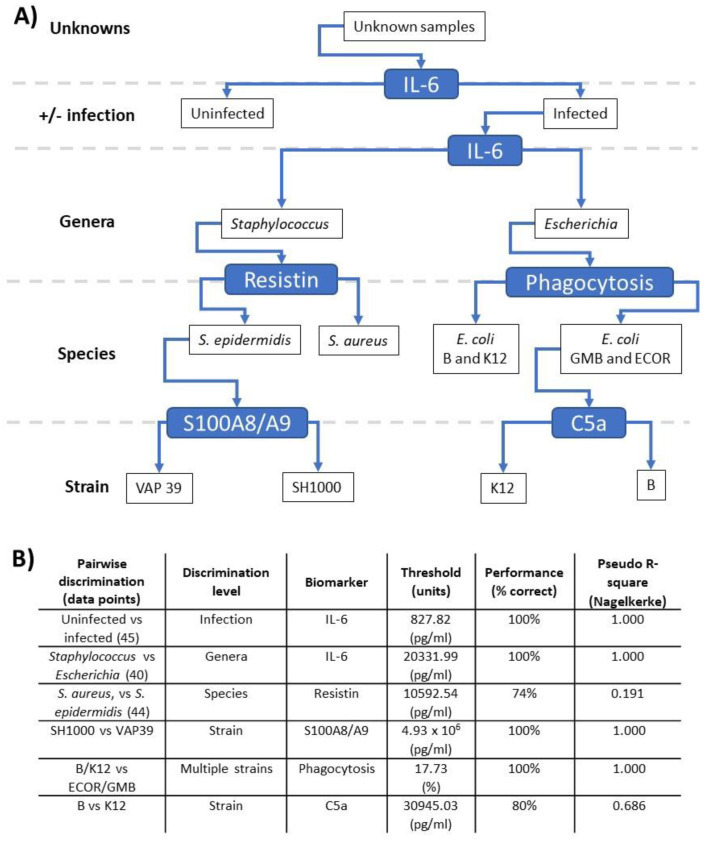
**Pairwise discrimination of bacterial immune responses.** Data generated in the ex vivo models were analysed, including the 10 inflammatory outputs over three timepoints, leading to a branching classification process across genera, species and strain levels. The branching process is illustrated in the upper panel (**A**), while the pairwise comparisons driving this process are listed in the lower panel (**B**) with the discriminating biomarker, the threshold of the relevant biomarker level and associated performance value are shown. Values lower than and higher than the threshold should ‘turn left’ and ‘right’, respectively, in panel (**A**).

**Table 1 biomedicines-12-00724-t001:** Linear discriminant analysis (LDA) of immune responses. Data generated in the ex vivo models were analysed, including the 10 inflammatory outputs over three timepoints. The table shows the recorded accuracy of immune responses to predict bacteria at the genera, species and strain levels. The underlying analysis reported in this table is shown in Appendix A.

Level	Correct Predictions (Observations)	Correct Prediction (%)
Strain	16/40	40
Species	36/40	90
Genera	38/40	95

**Table 2 biomedicines-12-00724-t002:** **Variables identified as important to bacterial discrimination**. Data generated in the ex vivo models were analysed, including the 10 inflammatory outputs over three timepoints. (**a**) confirms the immune variables predicting strain (phagocytosis, S100A8, IL-10). (**b**) confirms the immune variables predicting species (IL-6, phagocytosis, resistin and TF3). (**c**) confirms the immune variables predicting genera (IL-6, TF3 and MIP-3α).

**a: Variables Predicting Strain**			
**Biomarker**	**Wilks Lambda**	***p* Value Overall**	***p* Value Diff**
Phagocytosis	0.564	0.0064	0.0064
S100A8	0.396	0.0050	0.1053
IL-10	0.259	0.0017	0.0546
**b: Variables Predicting Species**			
**Biomarker**	**Wilks Lambda**	***p* Value Overall**	***p* Value Diff**
IL-6	0.313	0.0000	0.0000
Phagocytosis	0.193	0.0000	0.0187
Resistin	0.133	0.0000	0.0888
TF3	0.095	0.0000	0.1527
**c: Variables Predicting Genera**			
**Biomarker**	**Wilks Lambda**	***p* Value Overall**	***p* Value Diff**
IL-6	0.391	0.0000	0.0000
TF3	0.336	0.0000	0.0179
MIP3	0.286	0.0000	0.0166

**Table 3 biomedicines-12-00724-t003:** **Advantages and limitations of human ex vivo whole blood models**. List of considerations for implementation of whole blood models in bacterial discrimination and biomarker discovery.

Advantages	Limitations
Simple and rapid to perform	Lack of endothelial cells and other immune mediator-producing cells
Can measure both soluble and cellular mediators of inflammation	Presence of anticoagulants interfering with complement and coagulation pathways
Versatile application to a variety of assays and disease situations	Absence of liver and acute phase proteins
Only basic healthy volunteer ethics required	Potential for donor variation
Experiments completed in one day	Requires checks on plastic/vehicle activation
Use of a primary biological tissue/fluid/suspension	Longer experiments (>8 h) not feasible
Complements the human endotoxemia model without compromising patient safety	Impact of disease severity (sepsis vs. septic shock) not possible to measure

## Data Availability

Datasets presented in this manuscript are available upon request from the corresponding author.

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
