# Peer review of "Using the Traditional Ex Vivo Whole Blood Model to Discriminate Bacteria by Their Inducible Host Responses"

_biomedicines, 2024, doi:10.3390/biomedicines12040724_

Round 1
Reviewer 1 Report
Comments and Suggestions for Authors
Chick et al. reports a new model for identifying infection using whole blood from healthy donors. The authors show that the use of whole blood from healthy donors discriminates genera, species and strain of 8 types of bacteria.
Were the authors able to highlight discrimination if there is a co-infection? Can they include this eventuality in fugure 5A.
Concerning figure 5 A is what the authors can add ranges (concentration and phagocytosis index) in order to be able to refine their discrimination model.
What is the rationale for using a 0.2 MOI for all investigated bacteria? A MOI of 0.2 in whole blood is considered an infection of what grade?
The authors could add in Figures 1C, 2F, 3C, 3F, 4C and 4F the species of bacteria investigated.
Author Response
Attachment uploaded

Reviewer 2 Report
Comments and Suggestions for Authors
Dear authors,
Your manuscripts “Using the traditional ex vivo whole blood model to discriminate bacteria by their inducible host responses” describes the study of
whole blood models for bacterial discrimination, that is new methodology and could play an important role for clinical tests.
The manuscript is well-written, contains all necessary parts, the methods are adequate, and the conclusions are supported by data.
However, I have some remarks to be revised:
L40 – “LPS” – the full name should be provided for the first time.
Section 2.7. – the whole text in section is in italics – correct, please.
Section 3.3 – “The hormone resistin is associated with all infection and may differentiate Gram-positive bacteria.”
Section 3.4. “Neutrophils take up significantly more Gram-positive bacteria than Gram-negative bacteria”.
I can hardly agree that experiments on 1-2 representatives (species) of gram-positive/gram-negative bacteria can be approximated by the corresponding entire groups. I think you need to be more specific here, because you have not tested this blood parameter on other representatives of each group of bacteria.
Some references contain errors (see #11 and #12 for example – why do they contain inserts of German text?)
Author Response
Attachment uploaded
